# Pre-Operative Factors Associated with the Occurrence of Acute Kidney Injury in Patients Aged 65 Years and Over Undergoing Non-Ambulatory Non-Cardiac Surgery

**DOI:** 10.3390/healthcare10030558

**Published:** 2022-03-16

**Authors:** Wendy De Guglielmo, Jean Michel Rebibou, Serge Aho, Thomas Rogier, Gilles Nuemi, Claude Girard, Eric Steinmetz, Mathieu Legendre

**Affiliations:** 1Service de Néphrologie, CH Troyes, 10000 Troyes, France; 2Service de Néphrologie et Réanimation Métabolique, CHU F. Mitterrand Dijon, 21000 Dijon, France; jeanmichel.rebibou@chu-dijon.fr (J.M.R.); mathieu.legendre@chu-dijon.fr (M.L.); 3Service d’Epidémiologie et d’Hygiène Hospitalière, CHU F. Mitterrand Dijon, 21000 Dijon, France; serge.aho@chu-dijon.fr; 4Service de Médecine interne et Maladies de Systèmes, CHU F. Mitterrand Dijon, 21000 Dijon, France; thomas.rogier@chu-dijon.fr; 5Service DIM, CHU F. Mitterrand Dijon, 21000 Dijon, France; gilles.nuemi@chu-dijon.fr; 6Service d’Anesthésie Réanimation, CHU F. Mitterrand Dijon, 21000 Dijon, France; claude.girard@chu-dijon.fr; 7Service de Chirurgie Vasculaire, CHU F. Mitterrand Dijon, 21000 Dijon, France; eric.steinmetz@chu-dijon.fr

**Keywords:** pre-operative factors, acute kidney injury, surgery

## Abstract

This study sought to identify risk factors for acute kidney injury (AKI) from pre-operative variables in a population of subjects aged over 65. Eligible patients were aged 65 years or over who underwent scheduled non-cardiac, non-ambulatory surgery. Patients with a diagnosis of AKI recorded in the hospital’s databases were considered since cases, from which 300 patients with no diagnosis of AKI, were drawn at random as controls. In total, 81 cases of post-operative AKI and 239 controls were identified. The incidence of post-operative AKI was 2.87%. Pre-operative creatinine level (*p* = 0.0001), a history of respiratory insufficiency (*p* = 0.04), prior vascular surgery (*p* = 0.0001) and abdominal surgery (*p* = 0.03) were associated with an increased risk of AKI after surgery. These four variables calculated a score and developed a nomogram for predicting occurrence of post-operative AKI. A history of renal disease was associated with increased risk of post-operative AKI, predominantly in cases of vascular or abdominal surgery.

## 1. Introduction

Acute kidney injury (AKI) is a frequent complication in hospitalized patients and is multifactorial in origin. During surgery and the post-operative period, the risk of AKI increases with indications of AKI ranging from elevated creatinine levels to the need for hemodialysis [1]. According to Bellomo et al. [2], up to 40% of in-hospital AKI occurred in the period after surgery. Moreover, post-operative AKI increases mortality and length of stay (LOS) [3,4,5,6,7,8], as well as the risk of progression to chronic kidney disease (CKD) [9,10]. The risk of developing other post-operative complications such as infection or poor wound healing also rises [10,11].

Several studies investigating the incidence of AKI after surgery show that there is an increased risk for people over the age of 55 years [3,4,6,7,12,13,14]. Fortescue et al. [15] published a risk score for AKI after coronary artery bypass graft surgery, whereas Rueggeberg et al. [16] developed a risk stratification model for predicting acute renal failure in orthotopic liver transplantation recipients. Kheterpal et al. [14] identified several predictors of postoperative acute renal failure after non-cardiac surgery in adult patients with previously normal renal function. To date, however, no study has specifically investigated risk factors for post-operative AKI in patients aged 65 years and older.

In many countries, populations are increasingly aging, which may impact the healthcare ecosystem. In France, almost half of all surgical interventions are performed on patients over 65 years, and by 2050, one-third of the French population will be over 60 years old [17]. Furthermore, the number of comorbidities increases with older age [18,19,20], and the risk of AKI or of any etiology is also higher in older patients [6,7,14,21]. This study aimed to identify risk factors for AKI from pre-operative variables in a population of subjects aged over 65. Our hypothesis was that a post-operative AKI is likely to be related to certain clinical and/or biological factors as well as surgery type.

## 2. Methods

### 2.1. Study Design

A single-center case-control study was performed at the University Hospital of Dijon in France.

### 2.2. Inclusion Criteria

All patients aged 65 years and over, who underwent surgery during the study period from 1 January to 31 December 2015, were eligible for inclusion. Patient files were selected based on diagnostic codes entered into the hospital’s computer database.

### 2.3. Exclusion Criteria

Subjects undergoing emergency surgery, cardiac surgery, endovascular surgery, nephrectomy and ambulatory surgery (LOS < 24 h) were excluded. Patients with CKD receiving hemodialysis, patients with AKI diagnosed before surgery (defined as pre-operative serum creatinine significantly higher than previous documented values according to the Kidney Disease Improving Global Outcomes (KDIGO) 2012 classification and patients whose files had a high percentage of missing data were also excluded.

### 2.4. Definition of Cases

AKI was defined according to the KDIGO 2012 classification, comparing pre-operative serum creatinine levels (within three weeks prior to surgery) and those obtained within seven days after surgery. Estimated glomerular filtration rate (eGFR) was calculated using the Chronic Kidney Disease Epidemiology Collaboration (CKD-Epi) formula. Due to the high rate of missing data, diuresis criteria were not used to define AKI in this study. Recovery of renal function was defined as a reduction of at least 25% in creatinine levels during the in-hospital stay/follow-up, by a return to renal function that was compatible with weaning from hemodialysis, or by the return to pre-operative serum creatinine levels. Follow-up data for all patients (having the most recent serum creatinine levels available) were obtained to identify the number of patients still undergoing follow-up at three and five years later, as well as to calculate the proportion of patients who progressed to CKD or end-stage kidney disease.

### 2.5. Data Collection

Patients in the hospital’s computer database with a diagnostic code indicating AKI in addition to a surgical intervention between 1 January and 31 December 2015 were identified as cases. Controls were randomly selected from the database for those aged over 65 years who underwent surgery during the study period to those who had no diagnostic code for AKI. The flowchart of the study population is shown in Figure 1. A total of 300 controls were selected to ensure that there would be at least one control for each case. Cases and controls were not matched because this would have excluded the matching variable from subsequent statistical analysis. Potential matching variables such as age, sex and type of surgery being potential predictors of AKI were, however, included in the analysis. Controls were selected randomly using a specific computer program. The initial data extraction was performed using Cpage software (Cpage Company, Dijon, France).

Pre-operative data recorded for cases and controls included demographic and anthropometric data, comorbidities (cardiovascular and respiratory), usual treatment (renin–angiotensin–aldosterone system (RAAS) inhibitors, diuretics, beta-blockers, bronchodilators, proton pump inhibitors and statins), hemoglobin, serum creatinine levels and eGFR. LOS, in-hospital death and time of death (if applicable), use of iodine-based contrast medium within the five days prior to surgery, need for blood transfusion, and use of nephrotoxic antibiotics were also recorded.

### 2.6. Statistical Analysis

Comparisons were performed using the chi squared or Fisher’s exact test for categorical variables and the Kruskal–Wallis test for quantitative variables. A random forest model was used to select data from univariate analyses that had the greatest statistical weight. Logistic regression with a robust estimation of variance was used for multivariate analysis. The log–linearity of the logistic model was verified with fractional polynomials. Goodness of fit was evaluated by calibration (with the Hosmer–Lemeshow test) and via discrimination (area under the Receiver Operating Characteristic (ROC) curve, AUC). Internal validity was verified using the bootstrap method with 1000 replications. *p* value < 0.05 was considered statistically significant. The multivariate model was used to produce a score to estimate the risk of post-operative AKI. All analyses were performed using Stata Statistical Software version 15 (StataCorp LLC, College Station, TX, USA).

## 3. Results

### 3.1. Study Population

In total, 2819 met the eligibility criteria among 7735 surgical interventions that were performed in the center during the study period. The flow chart of the study population is presented in Figure 1. Up to 283 patients had AKI, of whom 79 met the inclusion criteria for cases. Among the 300 randomly selected controls, 239 met the eligibility for controls, and two patients had post-operative AKI that was not noted in the database. These patients were switched to the “case” group, and as a result, a total of 81 cases of AKI (2.87% of all surgeries) and 239 controls were used for analysis.

### 3.2. Characteristics and Outcomes

The baseline characteristics of the study population are shown in Table 1. Median age was 76 years old in both groups. There were significantly more males among the AKI cases (71.6% vs. 47.7% among controls, *p* < 0.0001). Median LOS (20 days for cases vs. 8 days for controls, *p* < 0.0001) and mortality at 6 months (42% vs. 2%, *p* < 0.0001) were significantly higher among cases. Among the 81 cases with AKI after surgery, 30 (37.04%) had Stage 1, 27 (33.33%) had Stage 2 and 24 (29.63%) had Stage 3 AKI according to the KDIGO 2012 classification. AKI occurred within 24 h of surgery in 50% of patients and within 72 h in 100% of patients. Seven patients (8.64%) required renal replacement therapy, of whom five (5/7, 71.43%) had Stage 3 AKI. At six months, 34 (42%) cases had died versus seven (2.9%) controls (*p* < 0.0001).

### 3.3. Univariate Analysis

Pre-surgery renal function was significantly impaired in the AKI cases with higher serum creatinine levels (median 116 µmol/L vs. 76 µmol/L; *p* = 0.0001) and lower eGFR (49 vs. 75; *p* = 0.0001) compared to controls (Table 2). AKI cases more frequently had a history of vascular disease (coronary artery or peripheral artery disease), respiratory insufficiency, diabetes (*p* = 0.011) and obesity (BMI > 30 kg/m²) (*p* = 0.017).

There were more tobacco smokers among AKI cases than among controls. Cases also had a more frequent history of respiratory diseases. RAAS inhibitors, diuretics and PPIs were more frequently used among cases. Surgery type, use of general anesthetic and use of iodine-based contrast medium within the previous five days were also significantly related with the risk of AKI by univariate analysis. Pre-operative serum creatinine levels had the strongest predictive power for post-operative AKI. Figure 2 shows the odds ratio (OR) for post-surgery AKI according to the pre-operative creatinine levels. When pre-operative creatinine levels exceeded 150 µmol/L, the OR exceeded one, which indicated risk increase. This corresponded to a score of five points on the risk score.

### 3.4. Multivariate Analysis

Using the random forest model, nine variables were selected for inclusion in the multivariate analysis: history of vascular disease (including coronary and peripheral artery), respiratory disease (including a history of respiratory insufficiency and/or use of bronchodilators), age, pre-operative serum creatinine levels, pre-operative hemoglobin, RAAS inhibitor use, statin use and surgery type (vascular and abdominal). The coefficients associated with each variable are shown in Table 3.

By multivariate analysis, four factors were found to be associated with an increased risk of post-operative AKI, namely, pre-operative serum creatinine (*p* = 0.0001), a history of respiratory disease (*p* = 0.04), abdominal surgery (*p* = 0.03) and vascular surgery (*p* = 0.001). Using the factors identified by multivariate analysis, a nomogram was drawn (Figure 3), indicating the number of points associated with each risk factor and the probability of post-operative AKI associated with each value across the range of the total score. Figure 4 presents the ROC curve produced by the internal validation procedure with an area under the curve of 0.82.

## 4. Discussion

In this study, AKI was reported in 2.87% of all post-surgeries but covered more than 30% for abdominal and vascular surgeries of patients over the age of 65 years.

There is currently no effective treatment for the prevention of AKI after surgery [22], although a certain number of nephroprotective measures may be implemented to reduce the burden of this major public health problem [8,10]. Since 2013, the International Society of Nephrology (ISN) has been implementing a program aimed at reducing or even eradicating deaths related to AKI with effective preventive measures that exist. However, to ensure such programs are successful, patients most at risk must first be identified and are therefore the most to likely benefit.

The main risk factor for the development of post-operative AKI is pre-existing renal disease, which reflects prior renal impairment as reported in numerous previous studies [3,6,7,12,13,14,23]. In our study, pre-operative serum creatinine levels yielded better statistical performance than eGFR using the CKD–Epi formula, whose validity in older persons has not been clearly demonstrated. Data about pre-existing renal injury, such as proteinuria or microalbuminuria levels, hematuria or usual blood pressure were, however, lacking.

Age is also a major risk factor identified in most studies, with a significant increase in the risk of AKI after the age of 55 years. This study, however, shows that beyond 55 to 65 years old, increasing age was not significantly associated with the occurrence of AKI after surgery. The oldest patients (over 85 years) who underwent surgery were probably in good general health (sufficient enough for surgery), and there were fewer patients aged over 85 than patients aged between 65 and 84.

Beyond 65 years of age, the number of comorbidities plays a more considerable role than chronological age. Ideally, geriatric assessment should be performed for potentially vulnerable patients. Indications for surgery should be evaluated on a case-by-case basis for dependent patients; however, this may be hard to organize in terms of logistics.

RAAS inhibitors are used in numerous indications in primary and secondary prevention of cardiovascular disease [24,25,26]. Unsurprisingly, there was a high rate of RAAS inhibitor use in our population sample. In related literature, RAAS inhibitor use has been reported to be a risk factor for post-operative AKI [13,24,27,28]. In this study, they were found to be associated with the occurrence of AKI under univariate analysis. However, sufficiently detailed data concerning interruption and/or discontinuation of these drugs prior to surgery were unavailable. Therefore, no conclusive correlation on their possible role in AKI could be drawn.

The etiology of AKI post-surgery is multifactorial. Commereuc et al. [28] previously described that in people with polypharmacy, the kidney’s capacity to adapt is overwhelmed by the hemodynamic stress that anesthesia and surgery together represent. The renal injury seems to be primarily tubular, and its origin is complex, associating hemodynamic mechanisms, toxic effects induced by anesthetic drugs, fluids used for volume expansion, and iodine-based contrast media. We observed this result and want to draw attention to the age threshold of 65 years old. At this age, respiratory comorbidities are more frequent, and they weaken kidney adaptation capacities.

The internal validation of the score that was developed using a Bootstrap technique with 1000 replications, average calibration (as reflected by the Hosmer–Lemeshow test), and an area under the ROC curve of 0.82 was adequate. The wider validation of this score is, however, likely to be difficult. Further research is required to validate this score.

Human resourcing and financial cost of AKI episodes occurring post-surgery are two topics worth considering in further research, even more in an aging population.

## 5. Conclusions

In this study, the focus on older patients undergoing surgery, above normal pre-operative serum creatinine levels, with a history of respiratory disease and abdominal surgery and/or vascular surgery were all associated with an increased risk of post-operative AKI. However, older age did not have an impact on risk of post-operative AKI. A nomogram that attributes points to each of these factors may yield a total score that predicts post-operative AKI and may allow for one to quickly trigger kidney protection for patients at risk.

## Figures and Tables

**Figure 1 healthcare-10-00558-f001:**
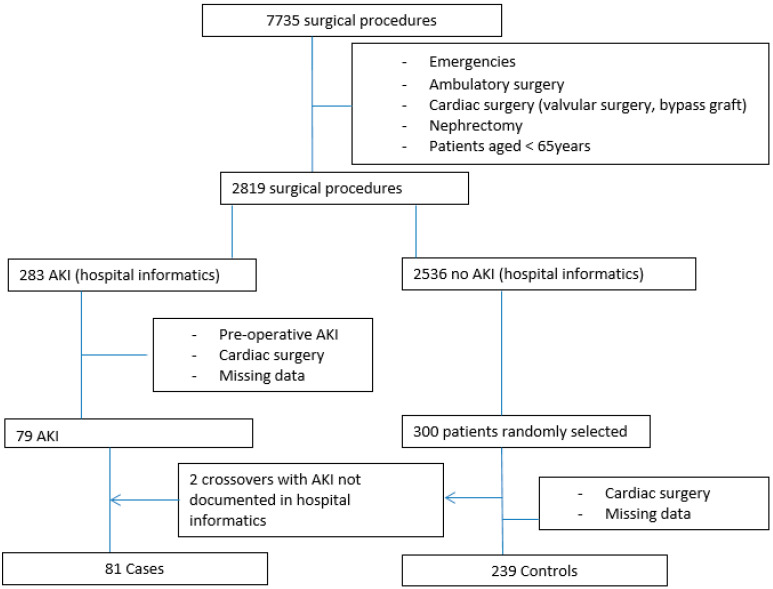
Flow chart of the study population that had acute kidney injury (AKI).

**Figure 2 healthcare-10-00558-f002:**
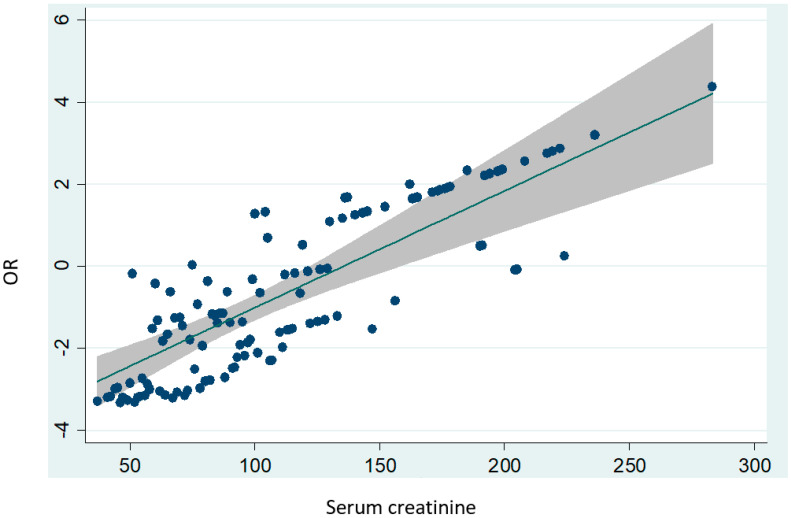
Odds ratios for post-operative acute kidney injury according to pre-operative serum creatinine levels.

**Figure 3 healthcare-10-00558-f003:**
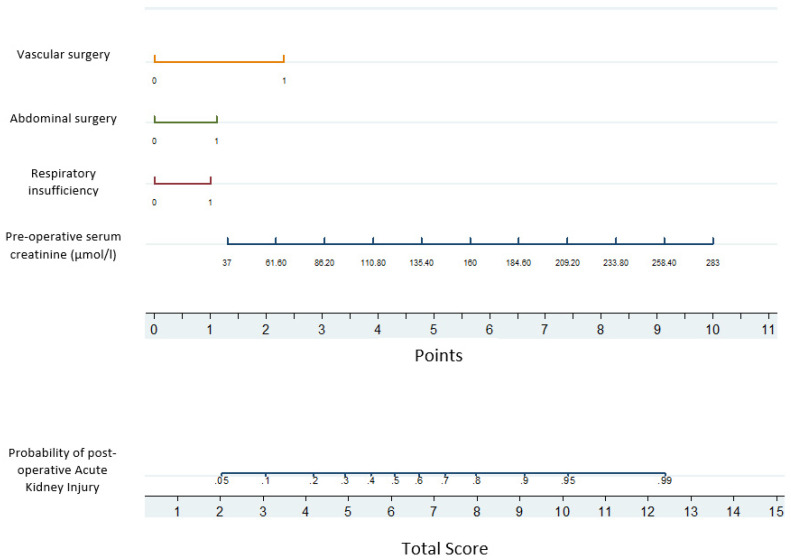
Nomogram indicating the number of points associated with each risk factor. Note: The number of points were found by drawing a vertical line from the value of the risk factor (0 = absent, 1 = present) to the X-axis. For example, if the patient had vascular surgery, the corresponding number of points is 2.5, and if creatinine was 110 µmol/L, then another four points was added to the patient’s score, yielding a total score of 6.5, and a probability of 50% experiencing post-operative AKI.

**Figure 4 healthcare-10-00558-f004:**
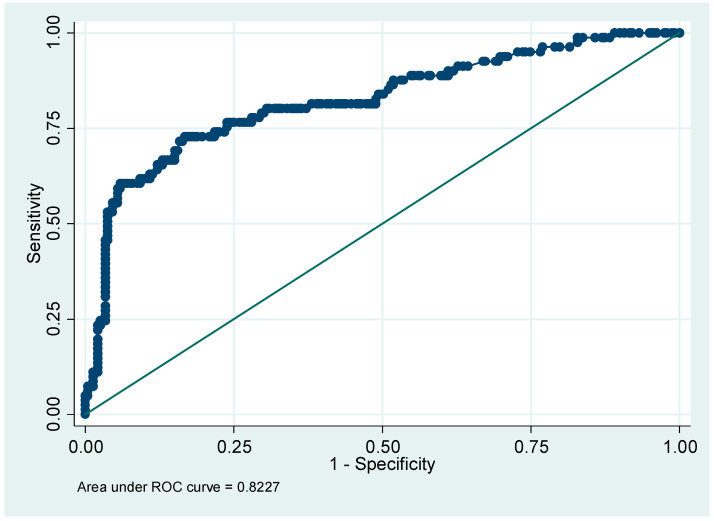
Receiving operating characteristic curve for the nomogram.

**Table 1 healthcare-10-00558-t001:** Baseline characteristics and outcomes among 81 cases with AKI and 239 controls.

	AKI Cases (N = 81)	Controls (N = 239)	*p* Value
**Sex** Females	23 (28.4)	124 (51.9)	
Males	58 (71.6)	115 (48.1)	<0.0001
**Age (median), years**	76	76	NS
**Length of stay (median, days)**	20	8	0.0001
**Death at 6 months**	34 (42.0)	7 (2.9)	<0.0001
**Time for surgery to death (median, days)**	10.5	21	0.18

**Table 2 healthcare-10-00558-t002:** Pre-operative characteristics of the study population.

		AKI (N = 81)	Controls (N = 239)	*p* Value
**Type of surgery**	Orthopedic	21 (26)	86 (36)	NS
	Abdominal	32 (39.5)	59 (24.6)	<0.0001
	Vascular	25 (30.8)	25 (10.5)	<0.0001
	Other	3 (3.7)	69 (28.9)	0.007
**General anesthetic**		81	220	0.003
**Biology**	Hemoglobin (g/dl)	11.8	12.5	NS
	Serum creatinine (µmol/L)	116	76	0.0001
	eGFR	49	75	0.0001
**Previous history**	Diabetes	25 (30.5)	41 (17.2)	0.011
	Obesity (BMI > 30 kg/m²)	28 (34.6)	51 (21.3)	0.017
	Hypertension	57 (70.2)	150 (62.8)	0.229
	Coronary artery disease	41 (50.6)	49 (20.5)	<0.0001
	Peripheral artery disease	35 (43.2)	52 (21.7)	<0.0001
	Respiratory insufficiency	30 (37)	41 (17.2)	<0.0001
**Smokers**	Current and former	35 (43.2)	76 (31.8)	0.043
**Treatment**	RAAS inhibitors	40 (49.4)	89 (37.2)	0.037
	Beta blockers	34 (42)	81 (33.9)	0.12
	Diuretics	40 (49.4)	70 (29.3)	0.001
	Statins	33 (40.7)	77 (32.2)	0.177
	PPI	35 (43.2)	71 (29.7)	0.019
	Bronchodilators	15 (18.5)	21 (8.8)	0.024
**Intra-abdominal**	n	35	73	0.027
	Laparoscopy	4 (11.4)	27 (37)	
	Laparotomy	31 (88.6)	46 (63)	0.003
**Use of iodinated contrast medium in the previous 5 days**	n	31 (38.3)	42 (17.6)	<0.0001
**Time since contrast use**	(median, days)	1	2.5	
**Nephrotoxic antibiotic therapy**	n	8 (9.9)	11 (13.6)	0.076
**Transfusion during surgery**	n	26 (32.1)	37 (15.5)	<0.0001

eGFR, estimated glomerular filtration rate; BMI, body mass index; RAAS, renin–angiotensin–aldosterone system; PPI, proton pump inhibitors.

**Table 3 healthcare-10-00558-t003:** Coefficients from the random forest model associated with each variable selected for inclusion in the multivariate analysis.

	Coefficient	SD	*p* Value	95%CI
**Age**	0.0136	0.0191	0.47	(−0.024–0.051)
**Pre-operative serum creatinine**	0.0243	0.0048	0.0001	(0.015–0.034)
**Vascular disease**	0.6243	0.3809	0.10	(−0.122–1.371)
**Respiratory disease**	0.7462	0.3649	0.04	(0.031–1.461)
**RAAS inhibitors**	−0.0513	0.3299	0.87	(−0.698–0.595)
**Statins**	−0.3815	0.3760	0.31	(−1.118–0.355)
**Abdominal surgery**	1.0361	0.4874	0.03	(0.080–1.991)
**Vascular surgery**	1.4843	0.4518	0.001	(0.059–2.369)
**Pre-operative hemoglobin**	−0.0211	0.0803	0.79	(−0.178–0.136)

SD, standard deviation; CI, confidence interval; RAAS, renin angiotensin aldosterone system.

## Data Availability

The data are available upon request to the corresponding author wendy.de-guglielmo@hcs-sante.fr.

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
