# Peer review of "Pre-Operative Factors Associated with the Occurrence of Acute Kidney Injury in Patients Aged 65 Years and Over Undergoing Non-Ambulatory Non-Cardiac Surgery"

_healthcare, 2022, doi:10.3390/healthcare10030558_

Round 1
Reviewer 1 Report
Comments to the Authors and Editor,
The present study focused on the risk factors for acute kidney injury (AKI) from pre-operative variables in a population of subjects aged over 65. For physicians, AKI after scheduled non-cardiac, non-ambulatory surgery should be burden. In addition, as authors described, no study has specifically investigated risk factors for post-operative AKI in patients aged 65 years and older yet. Thus, the aim and clinical points to be uncovered in the present study might be valuable and intriguing. Moreover, statistical analysis and presented data were simply analyzed, which led us to understand easily. However, several minor points should be addressed for acceptance. Please consider and response to following points which I pointed out.
Minor Points
1, Information in Table 4, indicating classification of AKI according to the KDIGO 2012 guidelines, is commonly known and recognized, which can be omitted from the manuscript.
- What do you mean 『NS? and P?』in Table 2? Please clarify the meaning of 『?』 in Table 2.
- In the comparison, between AKI group and controls group in Table 2, surgery time and the degree of metabolic acidosis (data for BE or HCO3, and so on) before surgery might be important factors that reflect the occurrence of AKI after surgery. Please show the data regarding operation time and presence of metabolic acidosis which might be cofounding factors for outcome in the present study.
Reviewer 2 Report
Postoperative kidney injury is a relevant prognostic factor after surgery. The authors try to solve this problem with a relatively large sample size. Analyzed patents are older than 65 years.
The end of the introduction should form a hypothesis. The authors want to prove or refute this with their analysis
Materials and Methods: What ICD codes did the authors use? Which computer program was used to select the controls (page 3)?
How was a significant p-value defined (page 4)?
What is meant by "coelioscopy"? (page 5)?
The discussion is very unfocused. The authors address relevant points from the literature. However, the reader misses the small reference to his own results. Both the final part of the discussion and the conclusion should clearly define which results are new compared to previous publications on the topic.
Basically, the topic is important. The new aspects of the analysis need to be better presented. What distinguishes this work from previous publications?
The reference section should be revised, since there are French terms included. Generally, English language should be improved. Th manuscript in the present form can not recommended for publication in HEALTHCARE
